

# Rapid measurement of RH-dependent aerosol hygroscopic growth using a humidity-controlled fast integrated mobility spectrometer (HFIMS)

Jiaoshi Zhang[1], Steven Spielman[2], Yang Wang[1,3], Guangjie Zheng[1], Xianda Gong[1], Susanne Hering[2], and Jian Wang[1]

[1]Center for Aerosol Science and Engineering, Washington University in St. Louis, St. Louis, Missouri, USA
[2]Aerosol Dynamics Inc, Berkeley, California, USA
[3]Department of Civil, Architectural and Environmental Engineering, Missouri University of Science and Technology, Rolla, Missouri, USA

*Correspondence to*: Jian Wang (jian@wustl.edu)

**Abstract.** The ability of aerosol particles to uptake water (hygroscopic growth) is an important determinant of aerosol optical properties and radiative effects. Aerosol hygroscopic growth is traditionally measured by humidified tandem differential mobility analyzers (HTDMA), in which size-selected dry particles are exposed to elevated relative humidity (RH), and the size distribution of humidified particles are subsequently measured using a scanning mobility particle sizer. As a scanning mobility particle sizer can measure only one particle size at a time, HTDMA measurements are time-consuming, and ambient measurements are often limited to a single RH level. Pinterich et al. (2017b) showed that fast measurements of aerosol hygroscopic growth are possible using a humidity-controlled fast integrated mobility spectrometer (HFIMS). In HFIMS, the size distribution of humidified particles is rapidly captured by a water-based fast integrated mobility spectrometer (WFIMS), leading to a factor of ~ 10 increase in measurement time resolution. In this study we present a prototype HFIMS that extends fast hygroscopic growth measurements to a wide range of atmospherically relevant RH values, allowing for more comprehensive characterizations of aerosol hygroscopic growth. A dual-channel humidifier consisting of two humidity conditioners in parallel is employed such that aerosol RH can be quickly stepped among different RH levels by sampling from alternating conditioners. The measurement sequence is also optimized to minimize the transition time between different particle sizes. The HFIMS is capable of measuring aerosol hygroscopic growth of six particle diameters under five RH levels ranging from 20% to 85% (30 separate measurements) every 25 min. The performance of this HFIMS is characterized and validated using laboratory-generated ammonium sulfate aerosol standards. Measurements of ambient aerosols are shown to demonstrate the capability of HFIMS to capture the rapid evolution of aerosol hygroscopic growth, and its dependence on both size and RH.



## 1 Introduction

Hygroscopic growth is a key determinant of the liquid water content, optical properties, and radiative effects of atmospheric aerosols (Tang and Munkelwitz, 1994; Pilinis et al., 1995; Swietlicki et al., 2008). Water uptake by particles increases light scattering, promotes heterogeneous chemical reactions, and is essential for the formation of cloud droplets (Wang et al., 2002; Wex et al., 2009; Surratt et al., 2010; George and Abbatt, 2010). Aerosol hygroscopic growth is mostly measured using a humidified tandem differential mobility analyzer (HTDMA) (Rader and McMurry, 1986; Swietlicki et al., 2008;

Duplissy et al., 2009; Massling et al., 2011; Lopez-Yglesias et al., 2014). In a HTDMA, dry and charged particles are first size-selected by a Differential Mobility Analyzer (DMA), then exposed to elevated relative humidity (RH) in a humidity conditioner. The size distribution of humidified particles, which provides the distribution of particle hygroscopic growth factor, is measured by a second DMA using the scanning mobility technique. Whereas in theory HTDMA can measure aerosol hygroscopic growth  at a variety of humidity values, practically, field measurements are mostly limited to a single

RH level because of (1) the time-consuming scanning mobility measurements and (2) the long transition time required for aerosol sample flow RH to stabilize after setpoint change (Santarpia et al., 2005). Typically, a measurement cycle for five different particle sizes at a single RH requires a minimum of 30 min (e.g., Cerully et al., 2011).

Particle hygroscopic growth is a function of RH and the hygroscopicity parameter (Petters and Kreidenweis, 2007), which depends on particle composition. Both RH and aerosol composition can exhibit strong temporal and spatial variabilities

(Tang et al., 2019). In addition, the hygroscopicity parameter can vary substantially with RH (Pajunoja et al., 2015; Rastak et al., 2017; Liu et al., 2018). Therefore, measurements under a wide range of RH are often needed to comprehensively characterize aerosol hygroscopic growth behavior in the atmosphere (Sorooshian et al., 2008; Hersey et al., 2009; Rastak et al., 2017). RH-dependent hygroscopic growth measurements can also help capture the phase transition of aerosols with different chemical compositions, i.e., inorganic salts exhibit deliquescent behavior while organics often do not (Tang and

Munkelwitz, 1994).

Several instruments have been developed to improve the measurement speed such that hygroscopic growth can be characterized at multiple RHs with sufficient time resolution (e.g., Stolzenburg et al., 1998; Leinert and Wiedensohler, 2008; Sorooshian et al., 2008). Recently, Pinterich et al. (2017b) showed that aerosol hygroscopic growth measurements can be significantly accelerated using a humidity-controlled fast integrated mobility spectrometer (HFIMS). In essence, HFIMS

replaces the 2nd DMA in traditional HTDMA with a water-based fast integrated mobility spectrometer (WFIMS), which captures the size distribution of humidified particles instantly (Pinterich et al., 2017b; Wang et al., 2019). In this study, we present a prototype HFIMS that extends the fast hygroscopic growth measurements to a wide range of RH conditions (20% to 85%). A newly designed dual-channel humidifier is employed to allow quick changes of the aerosol sample flow RH.  In addition, the measuring sequence is optimized to reduce the "dead time" during the transitions between different sizes of

classified particles. The performance of the HFIMS is characterized using laboratory-generated ammonium sulfate aerosol standards. Ambient aerosols in a St. Louis urban area were measured using the HFIMS, and the results are presented to



demonstrate the capability of the HFIMS for capturing the evolution of the hygroscopic growth factor distribution, and its dependence on particle size and RH.

## 2 Methods

### 65 2.1 Dual-channel HFIMS

One challenge for fast hygroscopic growth measurements at multiple RH levels is the lengthy time required for conventional humidity conditioners to stabilize following a change of the RH setpoint. The humidity conditioning of size classified particles is mostly achieved using a Nafion exchanger. Due to the relatively slow diffusion of water molecules through the Nafion membrane, it often takes a few minutes or more for the Nafion exchanger to reach a new RH setpoint. Lopez-
Yglesias et al. (2014) used a "membrane-less" diffusion-based humidifier to accelerate the transition between sample RH setpoints. In their HTDMA, it takes about 4 min for the system to stabilize for a 5% - 20% (absolute value) change in the RH setpoint. The HFIMS developed here employs a dual-channel humidifier that allows for a quick change of the sample aerosol RH (Fig 1). In the HFIMS, the sample aerosol is first dried to below 20% RH by a Nafion dryer (MD-110 series, Perma Pure LLC), brought to a steady-state charge distribution in a soft X-ray aerosol neutralizer (Model 3087, TSI Inc), then size-
classified by a DMA (Model 3081, TSI Inc.). The classified aerosol sample is subsequently introduced into a dual-channel humidifier that consists of two identical Nafion exchangers (MD-110 series, Perma Pure LLC) that condition the size-selected aerosol at two RH setpoints independently. The shell flows of the Nafion exchangers are generated by mixing humid air from a bubble humidifier and dry air. The bubble humidifier is slightly heated to compensate for evaporative cooling. The RH of the aerosol sample flow is controlled by a proportional-integral-derivative (PID) controller that varies the
flowrate ratio of humid air to dry air through a proportional solenoid valve (FSV12, Omega Engineering Inc) on the dry air line. The size distribution of humidified aerosols is rapidly captured by a WFIMS operated with a constant separator voltage (Pinterich et al., 2017a). The WFIMS combines the fast integrated mobility spectrometer (FIMS) developed by Kulkarni and Wang (2006b, 2006a) and the laminar flow water-based condensation methodology of Hering et al. (2014). As with conventional scanning mobility particle sizer (SMPS), WFIMS uses a drift tube to separate particles of differing electrical
mobility sizes. But instead of counting particles of one mobility size at a time, WFIMS counts all sizes of mobility separated particles at once. Its parallel plate mobility separator is followed by a condensational growth and imaging system. At the end of the separator, particles are distributed across the sheath flow, according to their mobilities. The entire flow continues into a condensational growth section that enlarges the particles while they continue along their laminar flow trajectories. At the end of the growth section a sheet of laser light illuminates the grown droplets, and this two-dimensional image is captured
using a digital camera (Fig. 1). The imaging system detects individual droplets and records their particle mobility-dependent position, from which the complete size distribution is derived. The dynamic range of WFIMS is roughly a factor of 10 in mobility, which enables it to detect growth factors from 0.8 to 2.4 at a single separator voltage. Here, the WFIMS sheath flow is generated by mixing particle-free dry air with filtered humid air from another slightly heated bubble humidifier. The





sheath flow RH is maintained at the same setpoint as that of the aerosol sample flow by adjusting the ratio of humid and dry

air flow rates, which are controlled individually by a mass flow controller (Fig. 1). The total flow rate of the humid and dry

air flows is 18.0 LPM, slightly above the WFIMS sheath flow rate of 14.9 LPM, and the excess is exhausted. The total flow

rate of WFIMS, i.e., the sum of sample and sheath flow rates, is controlled at 15.2 LPM by a critical orifice. The sample flow

rate is monitored and maintained at 0.3 LPM through adjusting the sheath flow rate using a proportional solenoid valve

(0248A, MKS Instruments) driven by a PID controller.

The sample flow RH is monitored by a probe (HMP 60, Vaisala Inc.) immediately downstream of each Nafion exchanger. A

higher sample flow rate of 0.6 LPM is employed for both Nafion exchangers to reduce particle diffusion loss and the

response time of the RH probe to the RH change inside the exchanger. Given the WFIMS sample flow rate of 0.3 LPM, the

extra 0.3 LPM is exhausted just upstream of the WFIMS sample flow inlet. Due to the very large sheath to sample flow ratio

(i.e., 14.9/0.3), RH inside the WFIMS is dominated by sheath flow RH, which is measured using a probe with high accuracy

(HMP 110, Vaisala Inc.). A chilled mirror hygrometer (OptiSonde, General Eastern Instruments) is also operated inline to

measure the dew point of WFIMS sheath flow for periodic calibration of the sheath flow RH probe. The aerosol flow rate of

the upstream DMA is dictated by the total flow rate of the dual-channel humidifier of 1.2 LPM. The DMA sheath flow rate

is maintained at 12 LPM through a closed loop to achieve a 10:1 sheath to aerosol flow ratio.

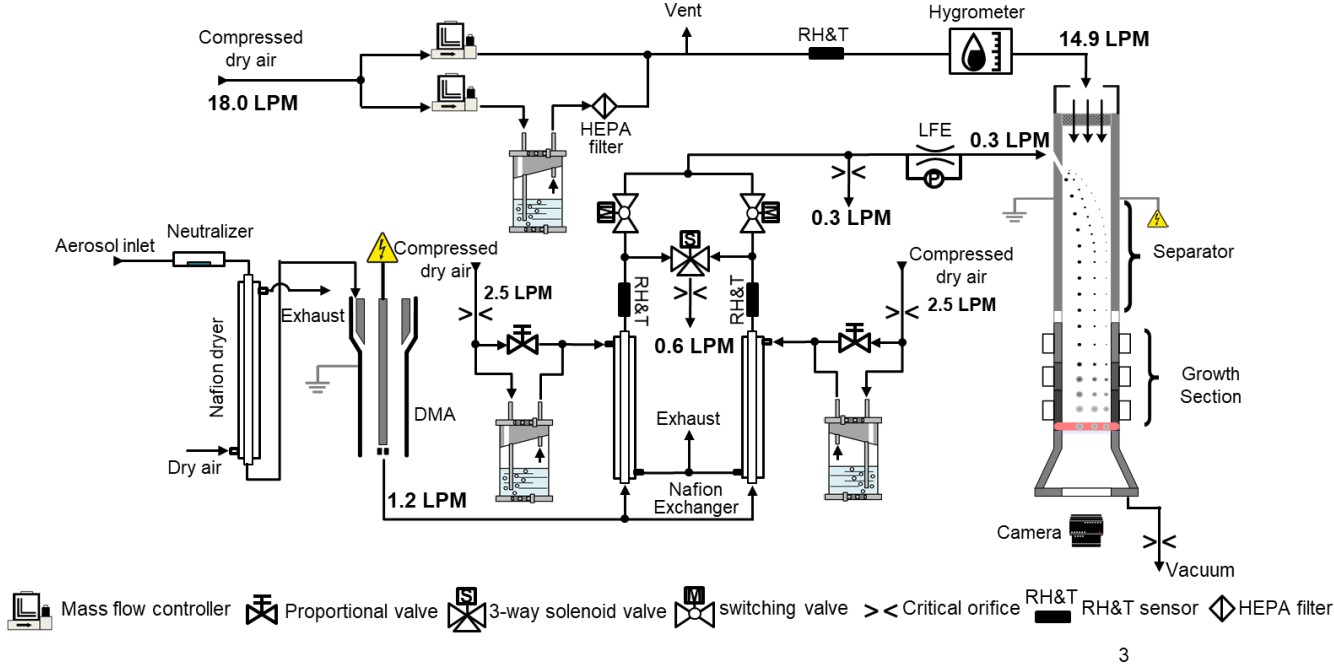


**Figure 1. Schematic diagrams of dual-channel humidity-controlled fast integrated mobility spectrometer.**





## 2.2 RH control and stepping

As WFIMS quickly captures the size distribution of humidified particles, the major obstacle for fast hygroscopic growth measurements at multiple RH levels is the long transition time (e.g., ~ 5 min) for the Nafion exchanger to stabilize following changes of the RH setpoint. In the new HFIMS, the transition time is minimized by using the dual-channel humidifier. The sample flow RHs of the two parallel Nafion exchangers are controlled independently. When the WFIMS samples aerosol conditioned by one Nafion exchanger, the other exchanger equilibrates to the next RH setpoint. The transition time between measurements at different RH levels is therefore minimized by sampling aerosols conditioned by the two exchangers alternately. With the dual-channel humidifier design, the transition time is now dictated by the stabilization of the WFIMS sheath flow RH, which is maintained at the same setpoint as that of the aerosol sample. The sheath flow RH is controlled by adjusting the mixing ratio of the humid and dry air flows, which are controlled by mass flow meters (Alicat Scientific) that have much faster responses. This approach leads to a faster control and stabilization of RH than in the Nafion exchanger. It requires less than 60 s for the sheath flow RH to be within 0.5% of the setpoint for a step change of 20% (absolute change).

## 2.3 Measuring sequence

Figure 2 shows the RH inside the WFIMS, DMA classifying voltage, and WFIMS separator voltage, during an example sequence for the hygroscopic growth measurements of ambient aerosols. The sequence includes five RH levels (20%, 40%, 60%, 75%, and 85%). At each RH level, hygroscopic growth is measured for aerosol classified at six dry sizes of 35, 50, 75, 110, 165, and 265 nm as recommended by the EUSAAR (European Supersites for Atmospheric Aerosol Research) project (Duplissy et al., 2009). Figure 2a shows the distribution of detected particle number ($R_i$) as a function of the normalized $x$-location ($\tilde{x}$, i.e., normalized distance traveled by the particles along the direction of electric field in the separator) in the viewing windows of WFIMS (Kulkarni and Wang, 2006a; Kulkarni and Wang, 2006b). Under the uniform electric field inside the WFIMS separator, larger particles travel a shorter distance in the direction of the electric field and are detected with lower $\tilde{x}$ values. In comparison, smaller particles travel further and are therefore detected at locations corresponding to higher $\tilde{x}$ values. At each of the 5 RH levels, the DMA HV is stepped through 6 voltages (288, 562, 1176, 2296, 4435, and 8980 V) to classify particles with diameter ranging from 35 to 265 nm (Fig. 2b). The WFIMS has a dynamic size range of a factor of 10 in electrical mobility (Kulkarni and Wang, 2006a), and its separator voltage is varied based on the DMA HV to cover the size range of humidified particles (i.e., 350, 650, 1400, 3000, 4500, and 5500 V). The upper limit of the WFIMS voltage is set at 5500 V to eliminate the possibility of arcing when operated at a high RH (i.e., 85%). The change of WFIMS voltage slightly lags (approximately 3 seconds) the DMA classifying voltage to offset the flow transport time between the outlet of the DMA and the exit of the WFIMS separator. The mean growth factor, i.e., the ratio of average humidified particle diameter measured by the WFIMS to the DMA-selected dry particle diameter, varies substantially among different particle sizes, indicating a strong size dependence of aerosol hygroscopic growth (Fig. 2c).

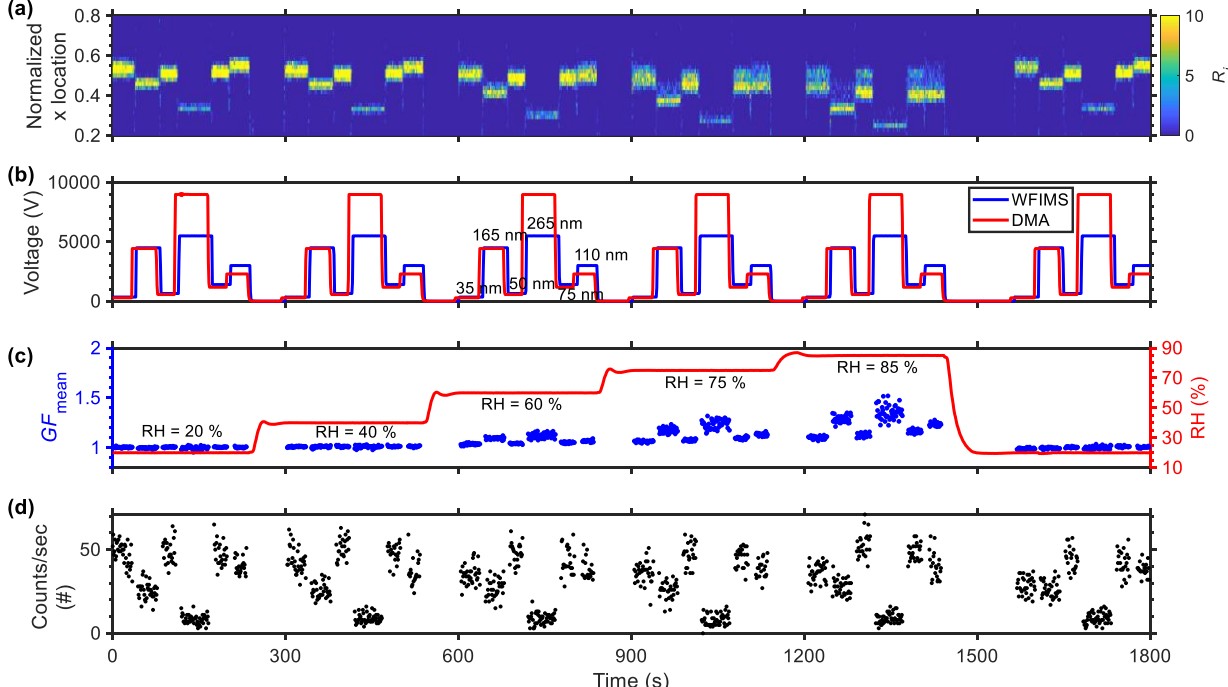

**Figure 2. Example of the hygroscopic growth measurement sequence of ambient aerosols with six dry sizes (35, 50, 75, 110, 165, and 265 nm) under five RHs (20, 40, 60, 75, and 85%). (a) The number of particles ($R_i$) detected by WFIMS as a function of normalized *x*-locations within the viewing window and time. (b)The DMA classifying voltage (red line) and WFIMS separator voltage (blue line). (c) RH inside the WFIMS separator and mean growth factor measured by HFIMS as a function of time. (d) Actual particle count rate of HFIMS as a function of time.**

The particle residence time inside the DMA and the time required for particles to travel from the DMA outlet to the WFIMS separator leads to "dead time" following changes in classified particle size. Given the parabolic flow profile in the tubing, particles travel with a range of velocities inside the HFIMS. As a result, following a change in the DMA classifying voltage, it takes some time for the vast majority of the particles classified at the previous size to exit the system. During this transition time, measurements may be strongly affected by particles classified at the previous size, especially when the particle concentration at the new size is much lower. Such transition time is common in measurements of size-resolved aerosol hygroscopicity. For example, in DASH-SP, it requires a maximum transition time of 17 s to switch between different DMA-selected sizes (Sorooshian et al., 2008). For traditional HTDMA, a transition time on the order of 20 s is negligible when compared to the long measurement time (i.e., several minutes). However, such transition time rivals the measurement time of HFIMS at each particle size and is one of the main obstacles for fast measurements at multiple sizes. Here the transition time is minimized by optimizing the size sequence as 35 -> 165 -> 50 -> 265 -> 75 -> 110 nm. At an RH of 85% or lower, the maximum range of growth factor (i.e., the ratio of humidified particle diameter to that of dry particles) for atmospheric aerosols is 0.8-2.0 (Gysel et al., 2007). For example, for dry particles of 35 nm, the diameter of humidified particles ranges from 28 to 105 nm. For the next dry size of 165 nm in the sequence, the possible size range of humidified particles is

between 132 and 495 nm. Because there is no overlap in the two size ranges of humidified particles, measurements at 165
nm are not affected by 35 nm particles remaining in the HFIMS, therefore the transition time between these two sizes (35 and 165 nm) can be significantly reduced to 5 seconds. In the optimized size sequence, an extended transition time of about 23 seconds is only required for the change of the classified size from 75 nm to 110 nm (i.e., the last step of the sequence). This relatively long transition time is necessary to ensure most of 75 nm particles have exited the HFIMS before the measurement at 110 nm commences. To balance the counting statistics at all six sizes, we determined the sample times based
on typical continental aerosol size distributions, and they are 35, 25, 25, 20, 35, and 50 s for 35, 50, 75, 110, 165, 265 nm, respectively. The sample times may be adjusted dynamically based on ambient aerosol size distributions. Figure 2d shows the particle counts detected by WFIMS per second. Because WFIMS measures aerosol size distribution at a time resolution of 1 second, sufficient particle counts (e.g., over 300 in 50 s for 265 nm) can be collected within the sample times for calculating the growth factor distribution (Wang et al., 2019). With this optimized size sequence, measurements of the
hygroscopic growth at a single RH for six particle sizes can be made every five minutes. In 25 min, which HTDMA typically requires to complete measurements of six particle sizes at a single RH, HFIMS can provide particle hygroscopic growth at 5 RH levels from 20% to 85% for all six particle sizes.

## 3 Results and discussion

### 3.1 Hygroscopic growth factor of ammonium sulfate particles

The capability of the HFIMS to accurately characterize particle hygroscopic growth was evaluated by measuring laboratory-generated ammonium sulfate particles, whose hygroscopic growth behavior has been well documented (Onasch et al., 1999; Martin, 2000). First, the sizing accuracy of WFIMS is confirmed by the excellent agreement between the mean diameter measured by WFIMS under dry conditions (i.e., RH = 20%) and DMA centroid diameter (Fig. 3). For six nominal sizes, i.e., 35, 50, 75, 110, 165, and 265 nm, the maximum difference between WFIMS measured mean diameter and DMA centroid
diameter is below 2%.

We further examine the accuracy of HFIMS measurements by comparing hygroscopic growth of ammonium sulfate particles measured by the HFIMS with theoretical values. Figure 4 shows the size-resolved hygroscopic growth factor (GF) as a function of RH for ammonium sulfate particles with diameters of 35, 50, 75, 110, 165, and 265 nm, respectively. The orange and purple solid lines represent theoretical predictions of the growth factor curve with and without Kelvin effect taken into
consideration, respectively (Biskos et al., 2006; Bezantakos et al., 2016).The growth factor curve predicted by Extended Aerosol Inorganic Model (E-AIM) (Clegg et al., 1998; Wexler and Clegg, 2002) is shown by the green lines, and it agrees well with those predicted by the model with Kelvin effect neglected. The HFIMS measured hygroscopic growth curves show that deliquescence started at ~79.8% RH,  in agreement with previous studies (Tang and Munkelwitz, 1993; Martin, 2000). There is an increase of 1% RH of deliquescence RH observed for smaller particles (i.e., 35 and 50 nm), and the increase is
likely due to increased equilibrium water vapor pressure over the surface of smaller droplets (i.e., Kelvin effect) (Hämeri et





al., 2000). Above the deliquescence point (i.e., RH> 80%), the hygroscopic growth measured by the HFIMS is in good agreement with those predicted by the model with Kelvin effect taken into account. Owing to Kelvin effect, particle hygroscopic growth decreases with decreasing dry particle size. For example, at 85% RH, the hygroscopic growth factor of 265 nm particles is around 1.57, and it decreases to around 1.50 for particles of 35 nm.

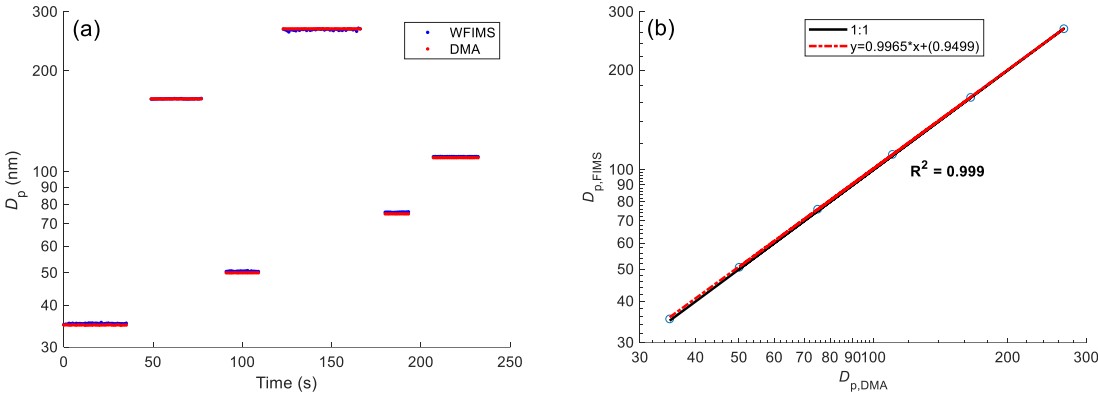


**Figure 3. (a) Time series of mean diameters measured by WFIMS at RH of 20% and DMA-selected sizes. (b) Scatter plot comparing DMA-selected particle size and WFIMS measured particle size and linear regression.**

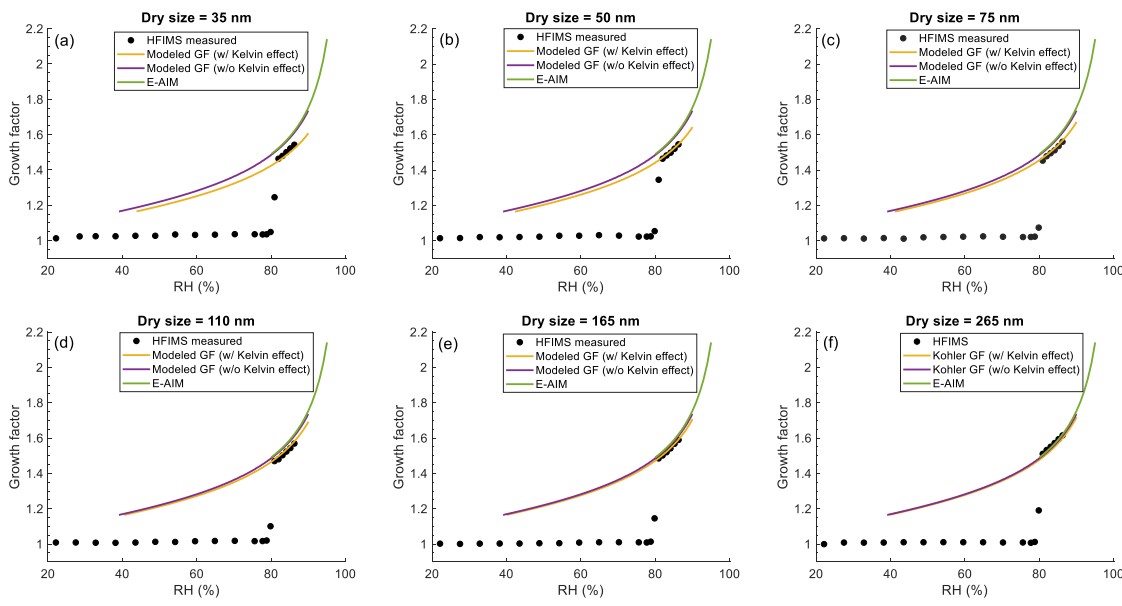


**Figure 4. Hygroscopic growth curves of ammonium sulfate particles for 6 DMA-selected sizes (35, 50, 75, 110, 165, and 265 nm). Black circles present growth factor derived from HFIMS measurements. Solid lines are model predictions of the growth factor curves with and without Kevin effects and calculated from E-AIM model.**



### 3.2 Hygroscopic growth measurement of ambient aerosols

### 3.2.1 High time resolution measurements of hygroscopic properties Ambient aerosol

Hygroscopic growth of ambient aerosols was measured by the HFIMS on the Danforth campus of Washington University in St. Louis, US (38° 38' N, 90° 18' W, 10 m a.s.l.), which is just beyond the western edge of the City of St. Louis. The measurement sequence is shown in Fig. 2. Aerosol size distribution ranging from 11.7 to 429.4 nm in particle diameter was measured concurrently by a SMPS ( model 3938, TSI Inc.). Figure 5a shows the evolution of aerosol size distribution during

a period of 3 days (March 1 - 4, 2021). Each particle size distribution was fitted by a sum of up to 3 lognormal modes. The time series of the fitted diameter of the dominate mode (i.e., the mode with the highest number concentration) is shown by black dots in Fig. 5a. Strong variations in both aerosol size and number concentration were evident during the sampling period. The high concentration of nucleation mode particles and the continuous increase of the mode diameter indicate a regional new particle formation (NPF) followed by condensational growth of newly formed particles on March 2[nd]. The NPF

event was observed at around 10:00 am local time (LT) and the newly formed particles continued to grow until the early morning of the next day (March 3[rd]).

The probability density function of the hygroscopic growth factor (GF-PDF) is retrieved from the HFIMS measurements using an inversion routine described in Wang et al. (2019). The high time resolution and size-resolved GF-PDF captures the variations of the aerosol hygroscopic growth and the mixing state with time, shedding light on the evolution of aerosol

particles. Figure 5b and 5d show the temporal variations of the GF-PDF at 85% RH for 35 and 265 nm particles, which represent those grown from newly formed particles and the pre-existing particles, respectively (Wu et al., 2016). The time series of mean GF under the five RHs (20%, 40%, 60%, 75%, and 85%) for 35 and 265 nm particles are shown in Fig. 5c and 5e. The mean GFs are essentially 1 at 20% and 40% RH for both 35 nm and 265 nm particles, indicating negligible water uptake in the low RH range. However, particles at the two sizes exhibit different hygroscopic growth at elevated RHs

and mixing states, especially during the NPF event. Particles with a diameter of 35 nm show a unimodal GF-PDF at 85% RH during most of the 3-day measurement period (Fig. 5b), suggesting an internal mixture. The mean GF of 35 nm particles at 85% RH remained low (i.e., ~ 1.1) before the NPF event, and it jumped from ~ 1.1 to ~ 1.25 at the beginning of the event (i.e., ~ 10:00 am LT on March 2[nd]). This GF increase is likely due to the participation of sulfuric acid in the formation and early growth of the new particles (Shantz et al., 2012; Wu et al., 2016). In comparison, the GF-PDF of 265 nm particles was

initially dominated by a single mode at ~ 1.5 while, gradually transitioned to bimodal as the NPF event proceeded, indicating a shift from internal mixtures to external mixtures. The bimodal GF-PDF consists of one mode with low GF (i.e., close to 1) and one with GF at ~ 1.5. The variation of GF-PDF suggests that the pre-existing aerosol changed from one that was dominated by aged particles with large contribution of inorganics (e.g., sulfate) to a mixture of both aged particles and freshly emitted ones that consisted mostly of organics with low hygroscopicity. As a result, the mean GF of 265 nm particles

decreased from 1.4 at the start of NPF event to 1.2 at ~18:00 pm LT on March 2[nd]. This transition is likely due to a change of the airmass sampled.



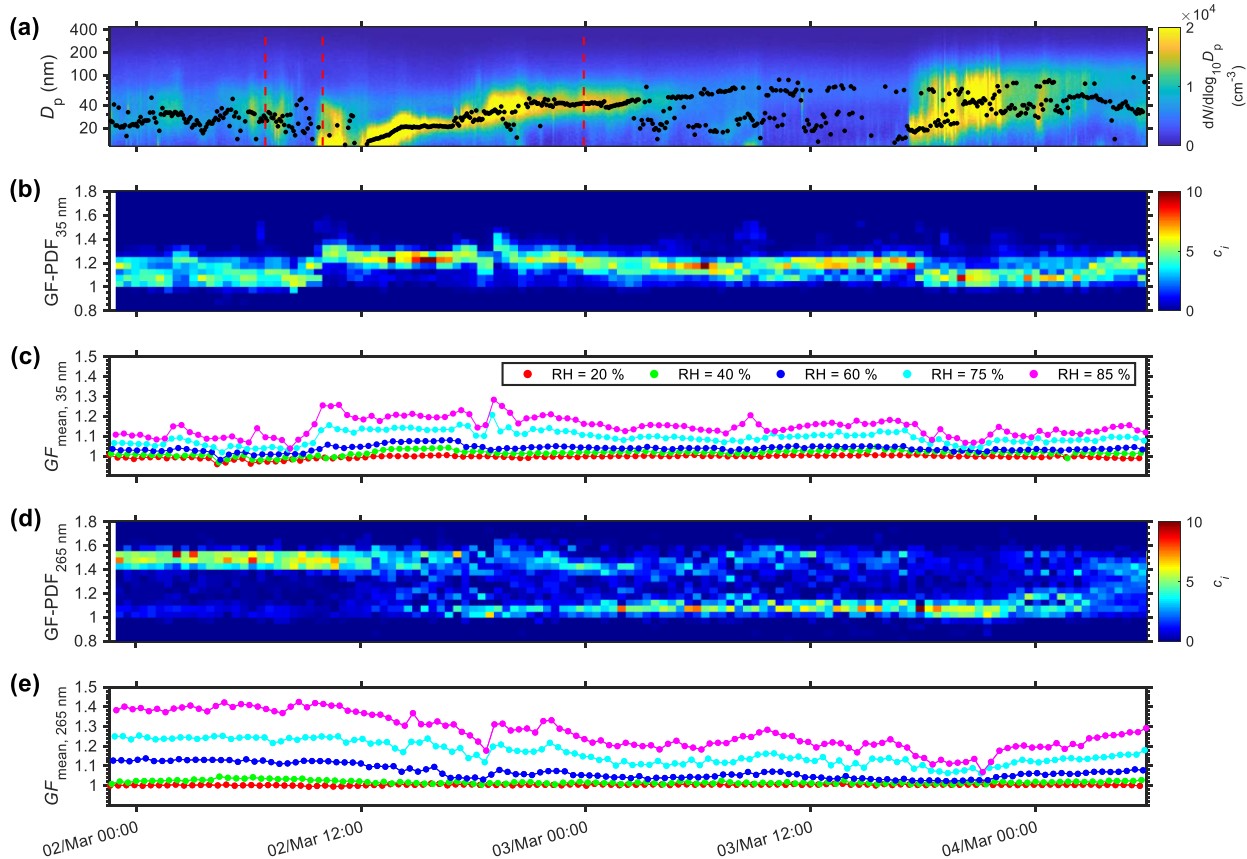

**Figure 5. Aerosol size distribution measured by SMPS and GF-PDF retrieved from HFIMS measurements on the Danforth campus of Washington University in St. Louis from March 1 to 4, 2021. (a) Temporal evolution of aerosol size distributions ranging from 11.7 to 429.4 nm. Black dots are the fitted geometric mean diameter of the mode with the highest number concentration. (b, d) Temporal evolution of probability density function of growth factor of 35 nm and 265 nm particles at 85% RH. (c, e) Timeseries of mean growth factor of 35 nm and 265 nm particles at five RHs levels (20%, 40%, 60%, 75%, and 85%).**

### 3.2.2 RH and size dependence of hygroscopicity distribution

At a given RH, particle hygroscopic growth depends on the hygroscopicity parameter (i.e., $\kappa$), which is a function of thermodynamic properties, including molar volume, activity coefficient, and surface activity, of the species within the particles (Petters and Kreidenweis, 2007). The probability density function of hygroscopicity, $\kappa$-PDF, is converted from GF-PDF based on the following relationship (Petters and Kreidenweis, 2007; Su et al., 2010; Liu et al., 2011):

$$\kappa(\text{GF}) = (\text{GF}^3 - 1) \cdot \left[ \frac{1}{\text{RH}} \exp\left( \frac{4\sigma_{s/a} M_w}{RT \rho_w D_p \text{GF}} \right) - 1 \right], \tag{1}$$





where $D_p$ is the diameter of dry particle and GF is the hygroscopic growth factor at different RHs measured by HFIMS. $\sigma_{s/a}$ is the surface tension of the solution/air interface, $M_w$ is the molecular weight of water, $R$ is the universal gas constant, $T$ is the absolute temperature, $\rho_w$ is the density of water.

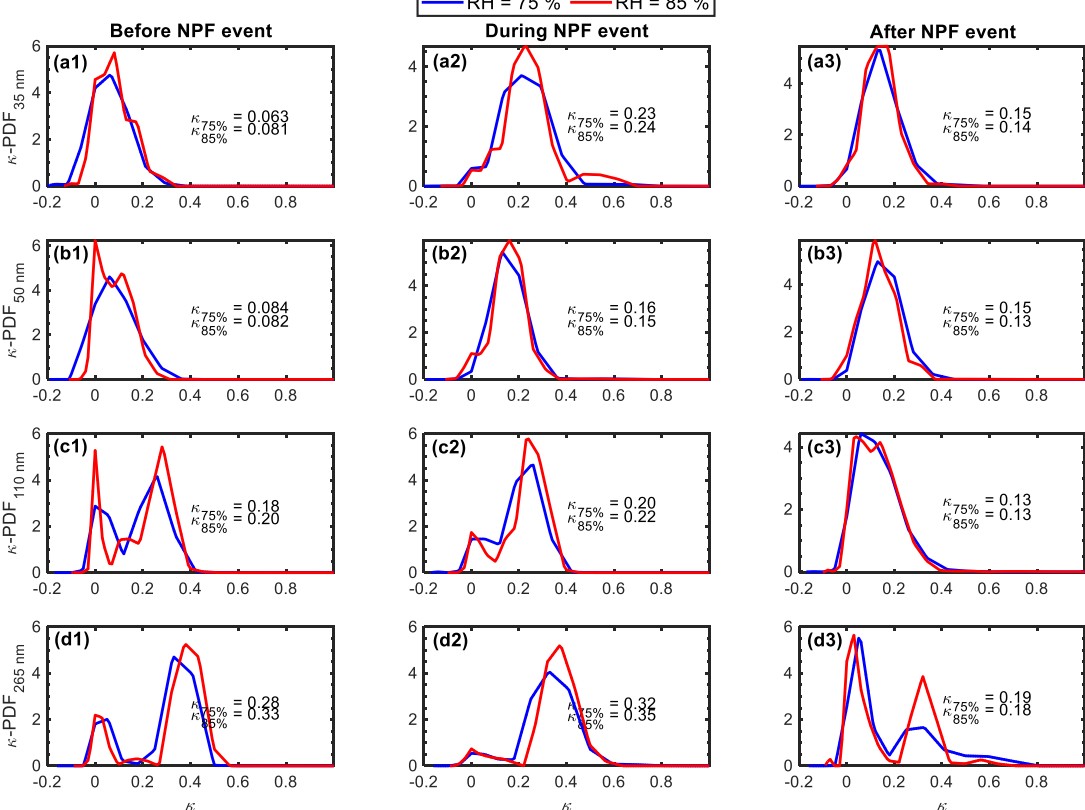

**Figure 6. Probability density function of hygroscopicity parameter (κ-PDF) before (a1-d1), during (a2-d2) and after (a3-d3) the NPF event period (06:53 am, 09:57 am, and 23:53 pm, respectively) on March 2$^{nd}$, 2021. The mean κ value derived from each κ-PDF is also given in the plots.**

Figure 6 shows the $\kappa$-PDF derived from GF-PDF at 75% and 85% RH before, during and after the NPF event, corresponding to 06:53, 09:57, and 23:53 LT on March 2$^{nd}$, as marked by red dashed lines in Fig. 5a. At all three timepoints, $\kappa$-PDF varies with particle diameter, indicating strong size dependence of particle chemical compositions. $\kappa$-PDF is unimodal for particles with diameters of 35 and 50 nm, showing these particles are internally mixed. The mean $\kappa$ values for 35 nm and 50 nm particles show substantial increases at the start of the NPF event compared to those prior to the event (e.g., mean $\kappa$ at 85% RH increased from 0.08 to 0.24 for 35 nm particles). This trend is consistent with the contribution of sulfuric acid to the formation and early growth of new particles (Kulmala et al., 2004; Wu et al., 2016). The mean $\kappa$ values for 35 nm and 50 nm particles then exhibit a slight decrease following the NPF event, possibly due to the depletion of sulfuric acid and increased contribution of secondary organics to the growth of nucleation mode particles in the late stage of the NPF event. Prior to and



at the beginning of the NPF event, the $\kappa$-PDF for particles of 265 nm was dominated by a mode with $\kappa$ value at ~ 0.4, consistent with the picture that pre-existing particles were dominated by aged particles with large contributions of inorganics (e.g., sulfate). The $\kappa$-PDF becomes bi-modal following the NPF events, suggesting a transition of the pre-existing particles to an external mixture of freshly emitted particles with low hygroscopicity (i.e., dominated by organics) and aged background

particles with higher hygroscopicity. The $\kappa$-PDFs derived from GF-PDF under two RHs (75% and 85%) mostly agreed well with each other, except for 265 nm particles prior to and at the start of the NPF, for which the dominant, more hygroscopic mode shows a slightly higher $\kappa$ value at 85% (Fig. 6d1-d2). This higher $\kappa$ at 85% RH may be due to an increase of the amount of water in the humidified particles that is available for the solvation of organics with low water solubility (Petters et al., 2009).

**4 Summary**

In this study, we present a humidity-controlled fast integrated mobility spectrometer (HFIMS) for fast measurements of aerosol hygroscopic growth. The HFIMS can measure the distributions of particle hygroscopic growth factor at six diameters (e.g., 35, 50, 75, 110, 165, and 265 nm, respectively) under five RH levels (e.g., 20, 40, 60, 75, and 85%, respectively) within 25 min. All 30 measurements are completed in less than 1 min per measurement, on average. The fast measurement

speed is achieved by combining the rapid measurements of aerosol size distribution by a water-based FIMS, the capability of a dual-channel humidifier for quickly stepping among different RH levels, and an optimized measurement sequence that minimizes the transition time between measurements at different particle sizes. Measurements of laboratory-generated ammonium sulfate aerosol standards show that the HFIMS is capable of accurately measuring aerosol hygroscopic growth. The capability of the HFIMS is also demonstrated by measuring ambient aerosols over a period of ~ 3 days. The HFIMS

measurements successfully capture the temporal variations and the size and RH dependences of aerosol hygroscopic growth, which provide insights into the evolution of atmospheric aerosol processes.

*Data availability*. Datasets related to this paper will be provided by the corresponding author (Jian Wang, jian@wustl.edu) upon request.

*Author contributions*. JW, SS, JZ and YW designed the instrument. JW and JZ designed the study. JZ carried out the experiments. JZ and JW prepared the manuscript with contributions from all co-authors.

*Competing interests*. The authors declare that they have no conflict of interest.

*Acknowledgements*. This work is supported by the U.S. Department of Energy's Small Business Innovation Research Program under contract DE-SC0013103 and Small Business Technology Transfer Program under contract DE-SC0006312.




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
