# Peer review of "Rapid measurement of RH-dependent aerosol hygroscopic growth using a humidity-controlled fast integrated mobility spectrometer (HFIMS)"

_Atmospheric Measurement Techniques, 2021_

## Referee Comment (RC1)

In this study, Zhang et al. developed a humidity-controlled fast integrated mobility spectrometer (HFIMS) for fast measurements of aerosol hygroscopic growth. Based on their previous work, a dual-channel humidifier and an optimized measurements sequence for different size particles were employed to achieve fast measurements. Compared to the H-TDMA techniques, the measurements time needed for a complete RH range cycle (20~85%) were largely shortened for the HFIMS, demonstrating its good feasibility in hygroscopic growth measurements for size-resolved ambient aerosols under different RHs. The manuscript is well written and easy to follow, I have several minor suggestions for authors' consideration.

Line 35: should be "an HTDMA".

Line 76: Please clarify the residence time of aerosols in the humidification section.

Line 95-96: Please clarify how to control the WFIMS sheath flow rate. A proportional solenoid valve coupled to a PID controller used here?

Section 2.2: RH in both aerosol and sheath flow was rather sensitive to the temperature fluctuation, please state how to maintain the stability of temperature during measurements.

Line 159-160: An optimized measuring sequence was introduced for six particle sizes; however, have the authors evaluated the effect of multi-charge on particle size determination, due to a large proportion of multi-charge particles (especially for 100-300 nm) selected by differential mobility analyzer (Shen et al., 2021, https://amt.copernicus.org/articles/14/1293/2021/)?

Line 212-214: Have the ambient aerosols been dried before measurement by a SMPS?

Section 3.2.2: I would like to draw the authors' attention to a recent review paper on the tropospheric aerosol hygroscopicity in China (Peng et al., 2020). It may be beneficial for the authors to discuss the RH and size dependence of ambient aerosol hygroscopicity. (https://acp.copernicus.org/articles/20/13877/2020/)

Line 269-270: Please add references to support this claim.

---

## Author Comment (AC1)

**Manuscript No.**: amt-2021-90

**Title**: Rapid measurement of RH-dependent aerosol hygroscopic growth using a humidity-controlled fast integrated mobility spectrometer (HFIMS)

5   We thank the anonymous referees for their valuable and constructive comments/suggestions on our manuscript. We have revised the manuscript accordingly and please find our point-to-point responses below.

**Comments by Anonymous Referee #1:**

*General Comments:*

10   *In this study, Zhang et al. developed a humidity-controlled fast integrated mobility spectrometer (HFIMS) for fast measurements of aerosol hygroscopic growth. Based on their previous work, a dual-channel humidifier and an optimized measurements sequence for different size particles were employed to achieve fast measurements. Compared to the H-TDMA techniques, the measurements time needed for a complete RH range cycle (20~85%) were largely shortened for the HFIMS, demonstrating its good feasibility in*

15   *hygroscopic growth measurements for size-resolved ambient aerosols under different RHs. The manuscript is well written and easy to follow, I have several minor suggestions for authors' consideration.*

*Detailed Comments:*

*Line 35: should be "an HTDMA".*

20   **Responses**: We have corrected it in the revised manuscript.

*Line 76: Please clarify the residence time of aerosols in the humidification section.*

**Responses**: The average transport time of particles from the inlet of the RH conditioner to the inlet of the WFIMS is ~ 3.2 s, and we have clarified this point in the revised manuscript in lines 82 – 83 as:

25   "The average transport time of aerosols in the humidification section, including the Nafion exchanger and the tubing leading to the WFIMS separator inlet, is ~ 3.2 s."

*Line 95-96: Please clarify how to control the WFIMS sheath flow rate. A proportional solenoid valve coupled to a PID controller used here?*

30   **Responses**: The total flow of WFIMS, i.e., the sum of the sample flow and sheath flow, is maintained by a critical orifice at a flow rate of 15.2 LPM. With the constant total flow, the desired aerosol flow rate of 0.3 LPM is achieved by PID control of sheath flow using a proportional solenoid valve. We have added a reference to detailed previous work in the revised manuscript in line 102.

35   *Section 2.2: RH in both aerosol and sheath flow was rather sensitive to the temperature fluctuation, please state how to maintain the stability of temperature during measurements.*

**Responses**: Yes, both the WFIMS separator and the growth section are temperature controlled, as well as the bubble humidifier. The temperature differences among different sections of aerosol and sheath flow passage, including WFIMS separator, are less than ~0.1℃, which corresponds to an RH variation of less than 0.6 % (relative change). Note that there were some variations of room temperature due to the activity in the lab, However, such variations are very slow in comparison, and they do not influence the growth factor measurements which take place at a much shorter time scale.

*Line 159-160: An optimized measuring sequence was introduced for six particle sizes; however, have the authors evaluated the effect of multi-charge on particle size determination, due to a large proportion of multi-charge particles (especially for 100-300 nm) selected by differential mobility analyzer (Shen et al., 2021, https://amt.copernicus.org/articles/14/1293/2021/)?*

**Responses**: We thank the reviewer for this constructive suggestion. The multi-charging effect is not taken into consideration in the current data inversion routine, and it will be a subject of our future work. We have clarified this in the revised manuscript in lines 229-231 as:

"The probability density function of the hygroscopic growth factor (GF-PDF) is retrieved from the HFIMS measurements using an inversion routine described in Wang et al. (2019). We note that particles carrying multiple charges could contribute substantially to the aerosols classified by the DMA, especially for particles larger than 100 nm (Shen et al. 2021). This multiple-charge effect is not currently accounted for in the inversion routine and will be a subject of future study."

*Line 212-214: Have the ambient aerosols been dried before measurement by a SMPS?*

**Responses**: Yes, the ambient aerosols were dried to an RH below 30%. It is now clarified in the revised manuscript in lines 219-220 as:

"Prior to the SMPS measurement, the aerosol sample was dried to an RH below 30% using a diffusion dryer."

*Section 3.2.2: I would like to draw the authors' attention to a recent review paper on the tropospheric aerosol hygroscopicity in China (Peng et al., 2020). It may be beneficial for the authors to discuss the RH and size dependence of ambient aerosol hygroscopicity. (https://acp.copernicus.org/articles/20/13877/2020/)*

**Responses**: We thank for the reviewer's suggestion. We have revised the text accordingly in lines 43-46:

"Particle hygroscopic growth is a function of RH and the hygroscopicity parameter (Petters and Kreidenweis, 2007), which is a function of particle composition and often varies strongly with particle size (e.g., Peng et al. 2020). Both RH and aerosol composition can exhibit strong temporal and spatial variabilities (Tang et al., 2019). In addition, the hygroscopicity parameter can vary substantially with RH (Pajunoja et al., 2015; Rastak et al., 2017; Liu et al., 2018; Peng et al., 2020)."

*Line 269-270: Please add references to support this claim.*

**Responses**: Following the reviewer's suggestion, we have added a reference, and the sentence now reads:

"The mean $\kappa$ values for 35 nm and 50 nm particles then exhibit a slight decrease following the NPF event, possibly due to the depletion of sulfuric acid and increased contribution of secondary organics to the growth of nucleation mode particles in the late stage of the NPF event (Dusek et al., 2010; Zheng et al., 2020)."

**5    References**

Dusek, U., Frank, G. P., Curtius, J., Drewnick, F., Schneider, J., Kürten, A., Rose, D., Andreae, M. O., Borrmann, S., and Pöschl, U.: Enhanced organic mass fraction and decreased hygroscopicity of cloud condensation nuclei (CCN) during new particle formation events, Geophysical Research Letters, 37, https://doi.org/10.1029/2009GL040930, 2010.

Zheng, G., Kuang, C., Uin, J., Watson, T., and Wang, J.: Large contribution of organics to condensational growth and formation of cloud condensation nuclei (CCN) in the remote marine boundary layer, Atmospheric Chemistry and Physics, 20, 12515-12525, 2020.

---

## Author Comment (AC2)

**Manuscript No.**: amt-2021-90

**Title**: Rapid measurement of RH-dependent aerosol hygroscopic growth using a humidity-controlled fast integrated mobility spectrometer (HFIMS)

5    We thank the anonymous referees for their valuable and constructive comments/suggestions on our manuscript. We have revised the manuscript accordingly and please find our point-to-point responses below.

**Comments by Anonymous Referee #2:**

*General Comments:*

10   *The manuscript by Zhang et al. entitled as 'Rapid measurement of RH-dependent aerosol hygroscopic growth using a humidity-controlled fast integrated mobility spectrometer (HFMIS)' introduces a novel technique to measure hygroscopic growth of atmospheric aerosol particles with high time resolution. The instrument is developed using the humidity-controlled fast integrated mobility spectrometer, which the authors have previously developed. The developed instrument is capable to measure hygroscopic growth of aerosol*

15   *particles with the high time resolution, which is difficult to be achieved by other techniques such as the HTDMA. The performance of the instrument was validated using ammonium sulfate and ambient particles in an urban area. The manuscript is well written. The topic is within a scope of the journal. The reviewer suggests accepting this manuscript after addressing the following minor comments.*

20   *Detailed Comments:*

*L57: 'hygroscopic growth measurements to a wide range of RH conditions (20% to 85%).'*

*HFIMS is capable to measure hygrscopicity of aerosol particles for the range of $20\% \leq RH \leq 85\%$. I was wondering if the instrument can be operated at RH > 85%, as higher RH is generally more important for investigating water uptake properties. It will be useful if there is a detailed information about limiting factors*

25   *for conducting measurement at elevated RH. Such information will help readers to think about future approaches in improving hygroscopicity measurements.*

**Responses**: We did not include 90 % RH as one of the setpoints mainly because (1) the main objective is to develop HFIMS for rapid measurements of aerosol hygroscopic growth at multiple RHs, and (2) it takes a

30   substantially longer time for aerosol flow RH to reach and stabilize at 90%. The control of aerosol RH at high values may be improved in future by employing longer Nafion exchangers with increased shell flow or membrane-less, diffusion-based humidifiers with high sheath flow rates.

35   *L70: 'Lopez-Yglesias et al. (2014) used a "membrane-less" diffusion-based humidifier to accelerate the transition between sample RH setpoints. In their HTDMA, it takes about 4 min for the system to stabilize for a 5% - 20% (absolute value) change in the RH setpoint.'*

*The idea of using two nafion tubing for rapid RH control is great. At the same time, I wondered the reason why the authors did not use the membrane-less diffusion-based humidifier if it is capable to change RH rapidly.*

5    **Responses**: On average, it takes about 5 min for aerosol RH to stabilize after a 20% (absolute value) setpoint change when using the Nafion exchanger. The "membrane-less" diffusion-based humidifier does have a faster response time. On the other hand, it takes about 4 min for the RH in the diffusion-based humidifier to equilibrate following a 20% step change in setpoint (Lopez-Yglesias et al., 2014), which is still too long for the rapid hygroscopic growth measurements at multiple RH levels. Therefore, we employed the dual-channel

10    humidifier (i.e., two Nafion exchangers operated in parallel) design such that the aerosol flows are conditioned at two different RHs (i.e., successive RH setpoints) simultaneously. The transition time between measurements at different RH levels is minimized by sampling aerosols conditioned by the two exchangers alternately. As it takes about 5 min to complete hygroscopic growth measurements at six sizes at one RH level, replacing the Nafion exchangers with the diffusion-based humidifiers in the dual channel design will

15    not further improve the time resolution of the measurements.

*L73: 'In the HFIMS, the sample aerosol is first dried to below 20% RH by a Nafion dryer'*
*Some organic aerosol particles that do not experience efflorescence might still retain measurable amount of water at the corresponding RH. It would be good if the authors could provide the reason why the criteria of*

20    *'below 20%' has been chosen.*
**Responses**: For field measurements, 20% RH is often used as the criteria for dry aerosol. We agree that at 20%, some organic aerosol particles that do not experience efflorescence might still retain a measurable amount of water. The RH of the aerosol sample should be kept as low as possible before being classified by the DMA in the HFIMS system, and this could be achieved by using a longer Nafion dryer with a higher

25    shell flow rate. On the other hand, in our measurements, particles exhibited essentially no growth when the RH increased to 40%, suggesting that the particles classified by the DMA contained a negligible amount of water (i.e., otherwise, we would expect a measurable increase in particle diameter at 40% RH).

30    *L91: 'The dynamic range of WFIMS is roughly a factor of 10 in mobility, which enables it to detect growth factors from 0.8 to 2.4 at a single separator voltage.'*
*I was wondering how the 'factor of 10 in mobility' in dynamic range could be translated to the variability in growth factor of 0.8 to 2.4 (probably because of large slip correction factors for smaller particles?). It would be better if this sentence could be rewritten in a clearer way.*

35

**Responses**: The dynamic range of WFIMS is roughly a factor of 10 in mobility. In the free-molecular regime, it corresponds to a factor of 3 in the size range due to the large slip correction factor (i.e., $Z_p \propto 1/D_p^2$ ). The

size range is even larger (i.e., more than a factor of 3) in the transition and continuum regimes. Therefore, at a single voltage, the dynamic size range of WFIMS is sufficient to cover the particle growth factors from 0.8 to 2.4. We have clarified this in the revised manuscript in lines 92 – 94 as:

"The dynamic range of WFIMS is roughly a factor of 10 in mobility, which translates into at least a factor of 3 in the particle diameter range. Therefore, the size range of WFIMS at a single voltage is sufficient to cover the growth factors from 0.8 to 2.4."

*L95: 'The total flow rate of the humid and dry air flows is 18.0 LPM, slightly above the WFIMS sheath flow rate of 14.9 LPM, and the excess is exhausted.'*

*Is there a reason why 18.0 Lpm of humidified flow needs to be prepared? I wondered why this approach was selected, rather than generating 14.9 lpm of humidified air directly using the mass flow controllers.*

**Responses**: This is designed to avoid pressure fluctuation inside the HFIMS system that could arise from any mismatch between the flow rates maintained by the mass flow controllers and the critical orifice. By excessing a small amount of flow into the ambient, we can ensure the pressure stability inside the WFIMS and thus particle sizing.

*L98: 'The sample flow rate is monitored and maintained at 0.3 LPM through adjusting the sheath flow rate using a proportional solenoid valve (0248A, MKS Instruments) driven by a PID controller.'*

*Could particle loss occur in the proportional valve?*

**Responses**: The proportional solenoid valve is used to adjust the sheath flow rate; therefore, no particle loss is expected to occur in the valve. The total flow of WFIMS, the sum of the sample flow and sheath flow, is maintained by a critical orifice at a flow rate of 15.2 LPM.  The aerosol flow is monitored by a laminar flow element and maintained at the desired 0.3 LPM via PID control of the sheath flow rate using the proportional solenoid valve. As the total flow is maintained at 15.2 LPM, this control strategy also ensures a sheath flow rate of 14.9 LPM.

*L122: 'which are controlled by mass flow meters'*
*Did the authors use mass flow controllers or mass flow meters?*

**Responses**: It should be mass flow controller. We have corrected it in the revised manuscript, and it now reads:

"The sheath flow RH is controlled by adjusting the mixing ratio of the humid and dry air flows, which are controlled by mass flow controllers (Alicat Scientific) that have much faster responses."

*L123: 'This approach leads to a faster control and stabilization of RH than in the Nafion exchanger.'*
*It was not clear why this approach provides faster response. Please clarify.*

**Responses**: In a Nafion exchanger, the sample flow RH is varied by transferring water molecules between the sample flow and shell flow through the Nafion membrane. The aerosol flow RH is therefore controlled by adjusting the RH of shell flow of the Nafion exchanger and it takes some time for the RH of aerosol flow inside the membrane tube to respond. As a result, the response of Nafion exchanger to RH setpoint change is substantially slower than directly varying the mixing ratio of dry and wet flows. The typical response time for the sheath flow stabilizing to a 20 % step change is less than 60 s by varying the mixing flow ratio. In comparison, for Nafion exchangers, it takes ~ 5 min for aerosol flow RH to stabilize following a 20% change in RH setpoint.

*L160: 'At an RH of 85% or lower, the maximum range of growth factor (i.e., the ratio of humidified particle diameter to that of dry particles) for atmospheric aerosols is 0.8-2.0 (Gysel et al., 2007). For example, for dry particles of 35 nm, the diameter of humidified particles ranges from 28 to 105 nm. For the next dry size of 165 nm in the sequence, the possible size range of humidified particles is between 132 and 495 nm.'*
*The range of 28 to 105 nm corresponds to hygroscopic growth factor of 0.8~3.0. similarly, the measurable range of hygroscopic growth for 165 nm (132 – 495 nm) corresponds to growth factor of 0.8~3.0. It would be better to explicitly mention the range of hygroscopic growth factors (rather than showing diameters) so that the readers can easily compare the measurement range for the present study with Gysel et al. (2007).*

**Responses**: The typical hygroscopic growth factor range of HFIMS is 0.8 ~ 2.4. And here we mistakenly stated it as 0.8 ~ 3.0 (although HFIMS can still make it for most of the sizes). We have corrected it in the revised manuscript (lines 163 - 168), and it now reads:

"Here the transition time is minimized by optimizing the size sequence as 35 -> 165 -> 50 -> 265 -> 75 -> 110 nm. At an RH of 85% or lower, the maximum range of growth factor (i.e., the ratio of humidified particle diameter to that of dry particles) for atmospheric aerosols is 0.8 to 2.0 (Gysel et al., 2007), within the minimum hygroscopic growth factor range (i.e., 0.8 to 2.4) of HFIMS. For example, for dry particles of 35 nm, the diameter of humidified particles ranges from 28 to 84 nm. For the next dry size of 165 nm in the sequence, the possible size range of humidified particles is between 132 and 396 nm."

*L239: 'The variation of GF-PDF suggests that the pre-existing aerosol changed from one that was dominated by aged particles with large contribution of inorganics (e.g., sulfate) to a mixture of both aged particles and freshly emitted ones that consisted mostly of organics with low hygroscopicity'*

*I wondered if the authors have any evidence to support the idea that freshly emitted less hygroscopic particles are mostly composed of organic species. There might have been some contributions of soot (or elemental/black carbon) particles (McMurry et al., 1996)*

**Responses**: We thank the reviewer for this point. Yes, freshly emitted less hygroscopic particles include possible contributions from soot as well. We have revised the sentence accordingly, and it now reads:
"The variation of GF-PDF suggests that the pre-existing aerosol changed from one that was dominated by aged particles with large contributions of inorganics (e.g., sulfate) to a mixture of both aged particles and freshly emitted ones that consisted mostly of low hygroscopicity particles (e.g., organic and soot particles)."

*Figure 6: Figure captions are overlapping with data for panels (d1 – d3).*

**Responses**: We have corrected it in the revised manuscript.

**References**

Lopez-Yglesias, X. F., Yeung, M. C., Dey, S. E., Brechtel, F. J., and Chan, C. K.: Performance Evaluation of the Brechtel Mfg. Humidified Tandem Differential Mobility Analyzer (BMI HTDMA) for Studying Hygroscopic Properties of Aerosol Particles, Aerosol Science and Technology, 48, 969-980, 10.1080/02786826.2014.952366, 2014.

McMurry, P. H., Litchy, M., Huang, P.-F., Cai, X., Turpin, B. J., Dick, W. D., & Hanson, A. (1996). Elemental composition and morphology of individual particles separated by size and hygroscopicity with the TDMA. Atmospheric Environment, 30(1), 101-108.